# INTERVENTIONAL BLACK-BOX EXPLANATIONS

## ABSTRACT

Deep Neural Networks (DNNs) are powerful systems able to freely evolve on their own from training data. However, like any highly parametrized mathematical model, capturing the explanation of any prediction of such models is rather difficult. We believe that there exist relevant mechanisms inside the structure of post-hoc DNNs that supports transparency and interpretability. To capture these mechanisms, we quantify the effects of parameters (pieces of knowledge) on models' predictions using the framework of causality. We introduce a general formalism of the causal diagram to express cause-effect relations inside the DNN's architecture. Then, we develop a novel algorithm to construct explanations of DNN's predictions using the *do*-operator. We call our method, Interventional Black-Box Explanations. On image classification tasks, we explain the behaviour of the model and extract visual explanations from the effects of the causal filters in convolution layers. We qualitatively demonstrate that our method captures more informative concepts compared to traditional attribution-based methods. Finally, we believe that our method is orthogonal to logic-based explanation methods and can be leveraged to improve their explanations.

## 1 INTRODUCTION

The design of deep neural networks (DNNs) is built on complex structure of neurons, layers and operations (e.g., convolutions, non-linearity and back-propagation). These biologically-inspired designs are able to evolve by their own from training data. Their high dimensional parameter space allows learning meaningful semantics from large data distributions and perform well on many tasks. However, makes difficult to capture explanation of their behaviour. This is one of fundamental obstacles for using these models on critical systems. Most popular explanation methods focus on creating saliency maps from classification models to visualize important features (Selvaraju et al. (2017b); Binder et al. (2016); Simonyan et al. (2014)) of a predicted class. However, saliency maps are not sufficient to reason on model behaviour and the explanations are not consistent between these methods. It is important to explain what are the mechanisms inside the hidden layers by which the model makes a prediction from an input. In this paper, we address this problem using causal inference.

Understanding of cause-effect relations in the DNN architecture is one way to make such black box models transparent and to explain their behaviour when tested on new data. Structural causal models (SCM) (Pearl (2009)) and their causal diagrams use interventions in terms of the *do*-calculus to express these relations. We rely on this framework to explain black box DNNs. We are interested in explaining post-hoc models, i.e., models after training. Recently, some work have addressed DNN explanations using causality (Chattopadhyay et al. (2019); O'Shaughnessy et al. (2020); Narendra et al. (2018)). These methods concentrate on the data generation process and input-output relations in classification models. Their goal is to explain the effects of changing aspects in input data on model predictions. Our work is different because we focus on the model itself, more precisely, the pre-trained knowledge stored in its parameters. We consider deep convolution models which are widely applied to computer vision tasks and have some applications in speech recognition and natural language processing. A neural network architecture is a form of Directed Acyclic Graph (DAG) models, in which neurons (nodes) are connected by directed edges form one layer to the next. A causal diagram or SCM was used to summarize the complex structure of DNNs (Chattopadhyay et al. (2019); Narendra et al. (2018)) such that interventions can be applied to explain the effect of a variable of interest on model prediction. The construction of the causal diagram depends on the

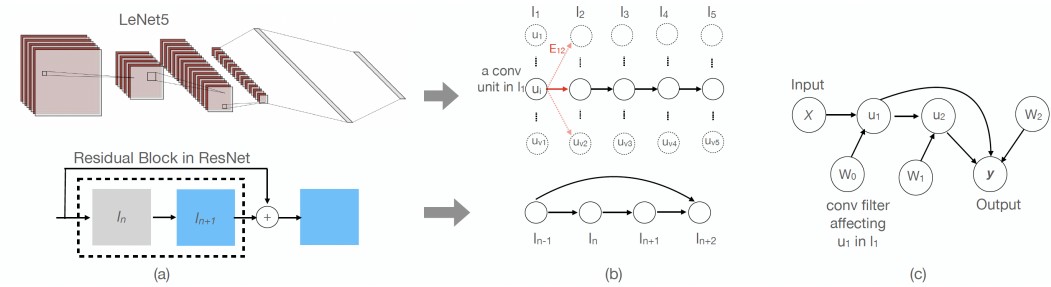

Figure 1: The proposed causal diagram of DNN model. (a) Examples of DNNs with convolution layers. (b) A graphical abstraction of the DNNs architectures to a simple DAG graph. Each node $u_i$ represents one channel in a hidden layer. (c) Illustrates the proposed causal diagram on an example of a CNN with 2 hidden layers, each composed of one convolution unit.

variables of interest that we would like to understand their effects. Some methods (Chattopadhyay et al. (2019); O'Shaughnessy et al. (2020); Harradon et al. (2018)) consider the latent features of a generative model as variables of interest. Narendra et al. (2018) focused on explaining the effect of the variance in convolution filters of a CNN. In this work, we provide a different view of the causal diagram (see Fig.1) which allows to perform causal reasoning on the entire structure of the DNN.

We summarize our contributions as follows. We build a causal graph of a post-hoc DNN. We develop an algorithm, termed interventional black box explanations, to find the causal mechanisms that explain the local and global DNN behaviour on individual samples and across samples, respectively. We show that our explanations can be used to correct or improve the probability of a prediction in test time. We consider, in this work, architectures for image classification. We capture explanations for LeNet and ResNet18 architectures using MNIST and ImageNet data inputs. We show that the explanations obtained by our method can be effectively used to remove noise in the model and improve its performance. Finally, like attribution-based methods, we provide visual explanations of classifier's behaviour computing visual concepts (semantics) from the effect variables (response of causal filters). We qualitatively demonstrate that our method captures more informative concepts strongly connected to model's prediction and useful for human interpretability.

## 2 INTERVENTIONAL BLACK BOX EXPLANATION

We begin our method by paving the way to the construction of the proposed causal diagram for post-hoc DNN (Section 2.1). In Section 2.2, we define the causal diagram followed describing of the proposed algorithm (Section 2.3).

### 2.1 FORMAL SETTING

We will use DNN and CNN interchangeably in this paper as we focus on classification networks comprising hidden convolution layers, but we keep in mind that our method is generic and can be applied on other architectures. In formal setting, a black-box CNN has two major components: a feature extraction module consisting of $n$ convolution layers followed by a classifier with $m$ fully connected layers. An input is an image $\boldsymbol{X} \in \mathbb{R}^{d_1 \times d_2 \times c_0}$, ($d_1, d_2$ are the spatial dimensions and $c_0$ is the channels number). The subscribe number indicates that this is the input layer $l_0$. An output $\boldsymbol{y} \in \mathbb{R}^K$ ($K$ is the number of classes) is a real-valued vector of predictions indicating the class logits of $\boldsymbol{X}$. The feature extraction module can be a simple structure of sequential layers $(l_1, ...., l_n)$ where each layer $l_i$ consists of $c_i$ convolution units (we call them nodes) and connected with all $c_{i+1}$ units in layer $l_{i+1}$. This structure can be more complex in state-of-the-art DNNs, such as ResNets, where skip connections are designed to jump over some layers (see Fig 1(a)). For simplicity, we assume non-linearity, batch normalization and pooling (if they exist) as parts of every node in layer $l_i$. A layer $l_i$ in the classifier module consists of $v_i$ neurons, and every neuron is densely connected to all $v_{i+1}$ neurons in layer $l_{i+1}$. The neurons are naturally nodes in those densely connected layers. We will substitute $c_i$ by $v_i$ to unify the notations. Consequently, we summarize the graphical structure of the DNN by $\mathcal{G}_M = (\{v_0, ..., v_n, ....v_N\}), \{E_{0,1}, ..., E_{N-1,N}\}$, with $N = n + m$ and $E_{i,i+1}$ is

edge vector connecting the nodes in layer $l_i$ to $l_{i+1}$ as shown in Fig 1(b). In post-hoc DNNs, each path communicates the information from layer $l_i$ to layer $l_{i+1}$ in a single direction starting from the input and passing forward to the output (prediction). It is an Acyclic Directed Graph (DAG). Note that this assumption is not valid on the network during training because of backpropagation. The pre-trained knowledge captured from training data is stored in the DNN's parameters (i.e., weights). We define $\boldsymbol{w}_i^k \in \mathbb{R}^{v_i}$ as the weight vector connected to the $k$-th node in layer $l_{i+1}$. We show in the next section the motivation for this notation.

## 2.2 THE CAUSAL DIAGRAM OF POST-HOC DNNs

A causal diagram is a graphical model that summarizes an existing knowledge where the nodes represent the variables of interest and edges represent the causal relationships between variables (Greenland & Pearl (2011)). A causal explanation consists of a causal diagram and symbolic queries defined by interventions, or *do-calculus*, to express cause-effect relations (Pearl & Mackenzie (2018)). For post-hoc DNNs, knowledge is encapsulated in its weights during the training phase. For CNNs, the filters of convolution layers express the pieces of knowledge that construct the entangled feature space and concepts of an input data. There are too many parameters to explore, and the goal of the causal graph is to uncover the important ones that explain the behaviour of the model. To explain the effects of these parameters on predictions, we propose a causal diagram as the example shown in Fig. 1(c). In this graph, we distinguish between two types of variables: parameters and features (network nodes). Our variables of interest are: parameters $\mathbf{W} = (\boldsymbol{W}_0, ..., \boldsymbol{W}_{N-1})$, test input $\boldsymbol{X}$ and the output (prediction logits) $\boldsymbol{y}$. In between, there are mediator variables (features) which transmit the effect of interventions on the parameters of intermediate layers to the output. As we can see, there is no direct effect of parameters variables in intermediate layers; there is only one direct effect on $\boldsymbol{y}$ which is the effect of $\boldsymbol{W}_{N-1}$ ($N$ is the logits layer). Also, there is no effect of input data $\boldsymbol{X}$ on parameters variables as they are independent in the case of post-hoc DNNs[1]. The parameters variables, $\boldsymbol{w}_i$, and features $u_i$ form a collider at $u_{i+1}$ in layer $l_{i+1}$ . This brings us to the following assumption which is the basic block of our proposed method.

**Assumption 1** *A de-confounded (robust) explanation of a black-box DNN can be defined by the measure of changes between the effect distributions $P(\boldsymbol{y}|do(\boldsymbol{w}_i^k), \boldsymbol{X})$ obtained by interventions on $\boldsymbol{w}_i^k$ and the observed distribution (outcome) $P(\boldsymbol{y}|\boldsymbol{X})$, for any $i \in \{1, ..., N\}$ and $k \in \{1, ..., v_{i+1}\}$.*

It is easy to check the validity of Assumption 1 from the causal graph shown in Fig. 1(c). First, the graph allows us isolating the parameters of interest and reason on the effect of every parameter on model output. Second, model parameters are not confounded by the skip connections existing in complex models, like ResNet. This is not the case when considering the features space where some variables may have common effects on multiple variables. We show such case in Fig. 1(c) where $u_1$ is a confounder ($u_2 \leftarrow u_1 \rightarrow \boldsymbol{y}$). Confounders require adjustment criterion (Pearl (2009)) to block any superiors correlations that give wrong effect estimates. This becomes more difficult in complex DNNs because of many confounders (Chattopadhyay et al. (2019)).

## 2.3 FINDING CAUSAL MECHANISMS IN POST-HOC DNN

We are interested in finding the mechanisms (causal pathways) along which changes flow from causal variables to the effects. These mechanisms will form explanations of the DNN's behaviour for every data input (local explanations) and across inputs (global explanations).

For simplicity, let us first consider the example in Fig 1 (c), where we have a single parameter $w$ in each layer. The causal model shows that the effect of a causal parameter $w_i^k$, or filter for convolution layers, on model output ($y$) is mediated by the effect of changes on variables (called mediators) along the paths from layers $l_{i+1}, ..., l_{N-1}$ after intervening on $w_i^k$. These changes will also transmit to the output variable in $l_N$. Generally, each layer has multiple neurons and we may apply interventions on a set of selected parameters ($k \in I$) to analyze their combined effect on model's output.

---

[1]The parameters update their values in the training phase using the training data inputs, which creates a causal path between data inputs and model parameters.

The interventions $do(w_i^k)$ imply changing the values of $w_i^k$. In convolution layers, each $w_i^k$ is a tiny set of pixels (e.g., 9 in case of $3 \times 3$ filter). Changing one single value would not carry out significant changes on the output. Therefore, we consider changing all the values of the filter. In fully connected layers, a parameter is a scalar value. The effect of changing one single parameter in the penultimate layer of the classifier would be highly significant to the output.

To find the causal mechanisms, we start by quantifying the direct effect on the prediction variable $y_j$ (for class $j$), which are in this case the parameters $\boldsymbol{w}_{N-1}^j$. We intervene on every single parameter and analyze its effect. Finding the direct effects enables identifying the causal paths in the network. However, for intermediate layers we cannot do interventions on every single parameter. This is computationally very expensive and not practical in inference time. Moreover, deep networks are highly parametrized, which would lead to a high variance in cause-effect relations thus making difficult to capture robust explanations. We propose a robust selection criterion to select the variables that we want to intervene on. This leads to the following proposition.

**Proposition 1** *(Intervention variables in intermediate layers) For $\boldsymbol{w}_i^K$, the set of indices $K \in \{1, ..., v_{i+1}\}$ of the mediators in intermediate layer $l_{i+1}$ and $I \in \{1, ..., v_i\}$ a subset of indices corresponding to the causal parameters $\boldsymbol{w}_{I,i}^K$. Then there exist causal paths between $u_{I,i}$ and $u_{K,i+1}$ and I will identify the intervention variables in $l_{i-1}$*

We proof Proposition 1 on simple example in Appendix A. As we notice, we measure changes in retrospective way, so the selection criterion is defined for layer $l_{i-1}$ after finding the causal parameters in layer $l_i$. To find the indices $I$ for layer $l_{i-1}$, we focus on changes which are statistically significant using a threshold $\delta_i$ which is defined as

$$\delta_i = (\mu_{\tilde{\boldsymbol{y}}} - \mu_j)/\sigma_{\tilde{\boldsymbol{y}}} \tag{1}$$

where $\mu_j$ is the original prediction (reference) of the true class $j$, and $\mu_{\tilde{\boldsymbol{y}}}$ is the average value of changed prediction and $\sigma_{\tilde{\boldsymbol{y}}}$ is the standard deviation. To capture the informativeness in causal parameters, we introduce in Proposition 2 the formula for the causal effect.

**Proposition 2** *(Causal effect) Given an input $\boldsymbol{X}$ and let $W_i^K$ be the set of causal variables (or paths) to the $K$ neurons in $l_{i+1}$, the causal effect $(CS_i)$ of $do(W_i^K)$ is a measure of the information flow from $W_i^K$ to $\boldsymbol{y}$ which is defined as:*

$$CS_i(n) = \mathbb{E}_{\boldsymbol{y}} \left[ \frac{\log p(\boldsymbol{y}|w_{n,i}^K = \alpha_n, \boldsymbol{X})}{\log p(\boldsymbol{y}|x_s)} \right] \tag{2}$$

*where $p(\boldsymbol{y}|\boldsymbol{X})$ is the Softmax of the prediction logits of an input $\boldsymbol{X}$, $n = (0, ..., v_i - 1)$, and $\alpha_n$ are the change values for each single parameter.*

We put details of Proposition 2 in Appendix A. The interventions that we used in this work are $\alpha_n = 0$. Propositions 1 and 2 uncover the causal mechanism of a hidden layer $i$ in the DNN's architecture. Algorithm 1 describes how we apply them to capture the causal mechanisms in all layers.

## 2.4 IMPLEMENTATION DETAILS

The implementation of our Algorithm requires an architecture $(\mathcal{G}_M)$ of the DNN, that can be a simple MLP or a CNN, a pre-trained knowledge (**W**), an input $\boldsymbol{X}$ and its prediction $\boldsymbol{y}$. The output of the algorithm is two dictionaries $G_+$ and $G_-$ which we use to explain the behaviour of the model given $\boldsymbol{X}$ and $\boldsymbol{y}$. The algorithm starts in a retrospective way from the penultimate layer $(l = N - 1)$ and the target class $K = j$. First, it computes the direct effect of the parameters on the output (prediction layer $l_N$). It searches two sets of indices: $I_+$ indicating the causal parameters, $\{\boldsymbol{w}_l^k\}$, which have positive effect, and $I_-$ that refers to the causal parameters, $\{\boldsymbol{w}_l^n\}$ ($n \neq k \; \forall k, n$), that have negative effect on the class of interest $y_j$, then updates $G_+$, $G_-$ respectively. Next, we use $I_+$ to update $K$ and select the intervention variables $\{\boldsymbol{w}_l^K\}$ for the next iteration (where $l$ becomes $l - 1$). We repeat the steps 3 to 10 until we end up with the input layer $l_0$.

---

**Algorithm 1** Causal mechanisms of post-hoc DNN

---

**Input:** $\boldsymbol{X}, \boldsymbol{y}, j, \mathcal{G}_M, N, \mathbf{W}$
**Output:** $G_+, G_-$
 1: Initialize $l = N - 1, G_+ = G_- = \{\}, K = \{j\}$
 2: **repeat**
 3:     $I_+ \leftarrow \{\}, I_- \leftarrow \{\}$
 4:     **for each** $w \in \boldsymbol{w}_l^K$ **do**
 5:         $w \leftarrow 0$ and compute $\tilde{\boldsymbol{y}}$
 6:         Add the index of $w$ to $I_+$ if $\tilde{\boldsymbol{y}} - \boldsymbol{y} < \delta_l$ else: Add the index of $w$ to $I_-$ if $\tilde{\boldsymbol{y}} - \boldsymbol{y} > \delta_l$
 7:         Compute causal effects $CS_l(n)$ in layer $l$
 8:         update $G_+$ and $G_-$ by cause-effect variables corresponding to $I_+$ and $I_-$
 9:     **end for**
10:     $l \leftarrow l - 1, K \leftarrow I_+,$
11: **until** $l = l_0$

---

## 3    EXPERIMENTS

We implement interventional black box explanations on deep neural networks trained for image classification tasks. We use two convolution neural network architectures: LeNet (Lecun et al. (1998)), a simple architecture composed of two convolution layers followed by two fully connected layers, and ResNet18 (He et al. (2016)). We capture post-hoc explanations for the model LeNet using MNIST data with 10 digits; and for ResNet18 using MNIST and a subset of ImageNet data (Deng et al. (2009)). For explainability, we selected the digits 3 and 8 from MNIST data, and some arbitrary classes from ImageNet.

### 3.1    CAUSAL EXPLANATIONS

**Per-sample causal mechanisms.** We implemented Algorithm 1 on single inputs to extract local explanations model's prediction. For this experiment, we chose an example where the model provided correct predictions and compute the causal effects $CS$ of interventions from each layer. Fig. 2 shows the causal effect computed over all layers (convolution and fully connected). This figure shows the differences between the causal mechanisms that the model use to reason on each different class. As we can see, the causal effects uncover different and common causal parameters between classes. To illustrate the behaviour of model's prediction, we show in Fig. 3 the effect distributions $\tilde{y}_3$ and $\tilde{y}_8$ and compare it to the prediction obtained using the original parameters of the model. For these examples, the original logit values are $y_3 = 8.19$ and $y_8 = 11.29$ and the probabilities obtained by the Softmax function are $93\%$ and $97\%$ respectively. Fig. 2 demonstrates an evidence on the presence of causal mechanisms that have either positive or negative effects on model prediction. Moreover, there are many parameters that don't have any effect on model behaviour, i.e. removing or keeping these parameters didn't change the output. As we mentioned in Section 2.3, removing these parameters is equivalent to removing the effect of the mediators (features) in the same layer.

**Cross-samples causal mechanisms.** We applied our algorithm on group of samples (100 positive example) to capture explanations about a global behaviour of the model. Fig. 4 shows results on MNIST digits predicted by LeNet. It illustrates the frequency of causal parameters in convolution and fully connected layers in terms of number of samples (colour and size of blobs are relative to the frequency). We also illustrate the causal filters which appeared in above $80\%$ of the samples. These filters explain the global patterns in each digit. The causal filters in the first convolution layer are identical across the samples of two classes. They are edge detectors and capture a global shape of the object. Note that their effect is not identical as we show later in Fig. 6. The causal filters in the second convolution layer are more interesting. They explain what makes a prediction to be a class 3 or 8 as their responses describe meaningful concepts. In fully connected layers, our method highlights different levels of importance of causal parameters. Those which persist for more than $80\%$ of samples are the most important and explain a global behaviour of the model. The causal parameters which have lower frequency explain local behaviour in small group of samples.

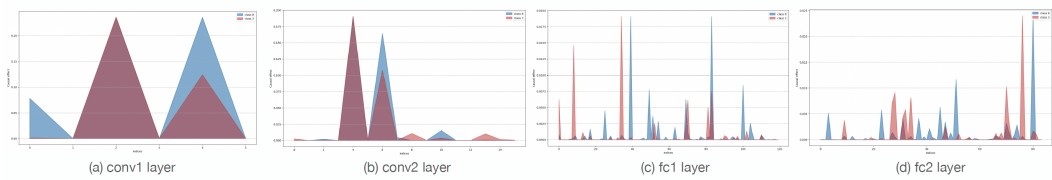

Figure 2: Causal effect of interventions on LeNet parameters over each layer for 3 and 8 digits using one test input

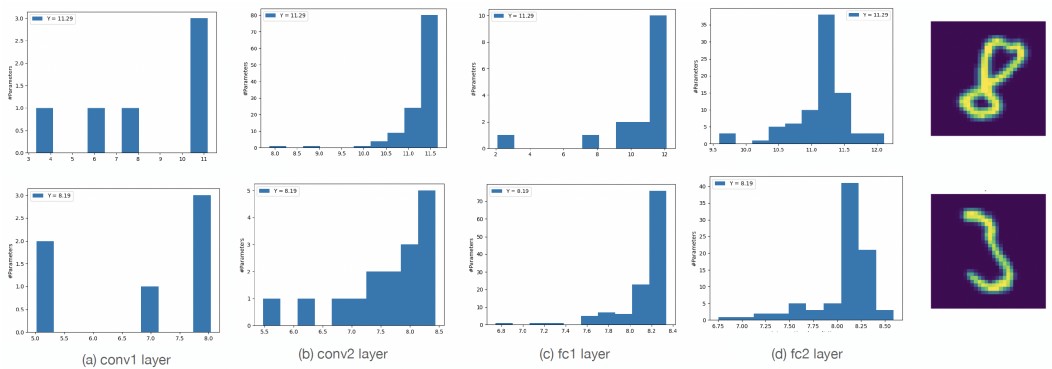

Figure 3: Distribution of effects of intervention variables over LeNet layers.

## 3.2 USE EXPLANATIONS TO CORRECT PREDICTIONS

In previous experiments, we provided explanations for positive predictions. In this experiment, we focus on negative samples to evaluate if our method can capture meaningful explanations to correct the predictions of wrong classes. We selected samples with digits 3 on which LeNet predicted the wrong digit 8. The model made wrong predictions for 11 samples from the test set. We implemented our algorithm on all negative samples belonging to this subset and extracted cause-effect relations corresponding to the true class. Fig 5 shows the causal parameters for the last convolution layer and the fully connected layers. Comparing the results to the group of 100 positive samples shown in Fig. 4 for digit 3, the algorithm consistently captures similar explanations corresponding to the strongest causal parameters. Other causal parameters are mostly related to local differences specific to each sample. This demonstrates that the extracted causes can be used to correct the model. To do so, we turned off (put to zero) all the non-causal parameters and the ones in $G_-$ and only kept the causal parameters in $G_+$. Those which are expected to have positive effect on the model. Then, we tested the model again on the data. As we show in Fig 5, the model is $100\%$ certain of its prediction with significantly high logit value compared to the original predictions where the average probability was $15\%$ for the true class. We show in Appendix B additional results on ResNet18.

## 3.3 COMPARISON TO EXPLANATION METHODS

Besides that our method captures the causal parameters and filters that affect DNN's prediction on each class, we can visualize the features in the mediators located in the paths of the causal graphs. These mediators are simply the neurons or channels (in case of convolution layers) that mediate the effect of causal parameters on model predictions. Based on Assumption **??**, we can find the mediators from the index set $I_+$ of the causal weights. For instance, the mediator in the last convolution layer of ResNet18, $layer_{4.1}.conv2$, is a 2D feature vector of dimension $K = 512$ with the $n$ effective channels where $n \in I_+$. Remaining channels are omitted. We obtain the attribution map, by computing the average at each pixel. This allows us comparing our method with other explanation methods, where the goal is to explore if the causal mechanisms lead to extract better explanations. We compared the visual explanations generated by our method with the ones generated by other explanation methods such as: Saliency (Simonyan et al. (2014)), Occlus (Zeiler

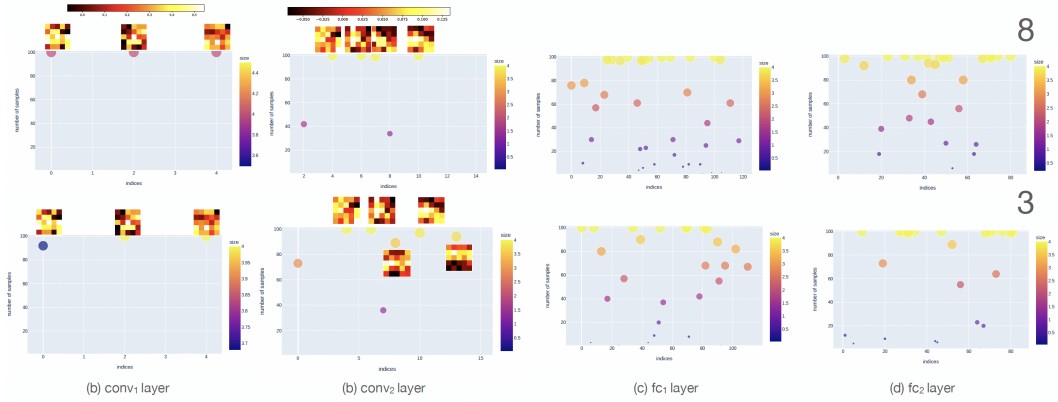

Figure 4: Global explanations and comparison between digits 3 and 8 using 100 samples. (a) and (b) show the frequency of causal filters causal filters in conv1 and conv2 layers, respectively. (c) and (d) show the frequency of causal parameters in fully connected layers. The x-axis indicate the indices of causal parameters.

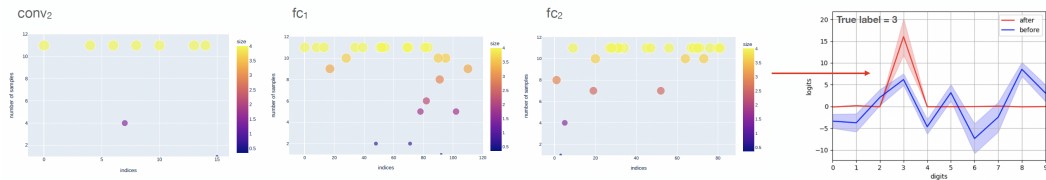

Figure 5: Causal parameters corresponding to the true class for LeNet when tested on negative samples. The figure on the right shows the original predictions of the model (*before*) and the corrected one

& Fergus (2014)), IG (Sundararajan et al. (2017)) and DeepLift (**??**). We also wanted to include the popular LRP (Binder et al. (2016)) method, however, this method doesn't work on complex architectures such as ResNet because the method doesn't handle the skip connections. Figure 6 shows results on MNIST digits using LeNet model. Comparing the explanations generated for digits 3 and 8, we can see that other methods don't provide informative explanations to recognize the digit 3 from 8. Our method shows the effects of the causal filters which explain the difference between digits through their shapes. In Fig. 7 we show causal explanations of ResNet18 model on arbitrary classes of ImageNet. Our method captures visual explanations from all layers. The causal filters capture different meaningful concepts (semantics) at multiple levels of hierarchy. We show here results from some layers and illustrate further details in Appendix C.

## 4 RELATED WORK

The work on explaining deep neural networks keep rising by the machine learning community because of the difficulties of computing robust explanations from these black box models. Many methods have been proposed by the community. In this paper, we focus on explanation methods of post-hoc models which are related to our research problem.

**Local and Global Explanations.** The main forms of explanations split into these two categories (Doshi-Velez & Kim (2017); Das & Rad (2020)). Global explanations aim at explaining and summarizing the behaviour of the model, for instance the mechanism underlying a classification model, over a group of instances (data points). Local explanations focus on a local data point under scrutiny. They capture explanations from a single instance and search the parts of input data that are most relevant for model's prediction. Rule-based methods and decision trees have been used to capture global explanations approximating the DNN models (Letham et al. (2015); Craven & Shavlik (1995)). These techniques are also used to capture local explanations by approximating the de-

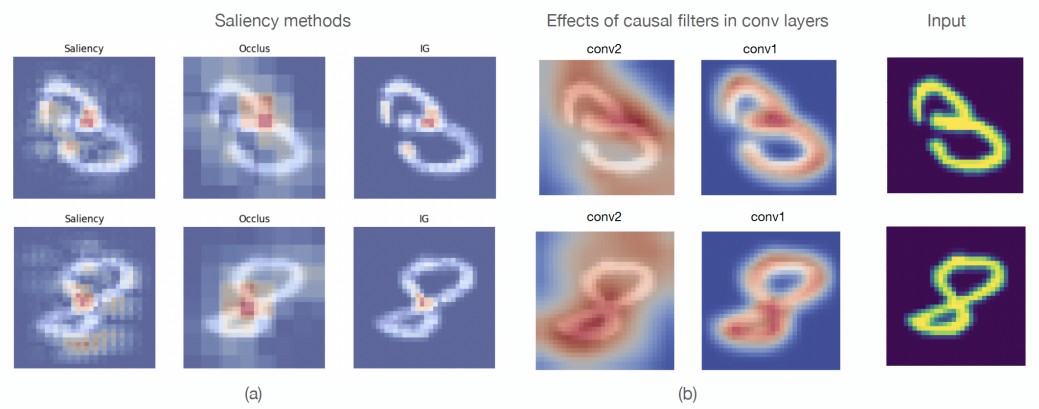

Figure 6: Visual attributions from different methods. Comparison of visual explanations between 3 and 8 digits. Our method generates visual explanations by integrating the response of the mediators corresponding to the causal filters in each conv layer.

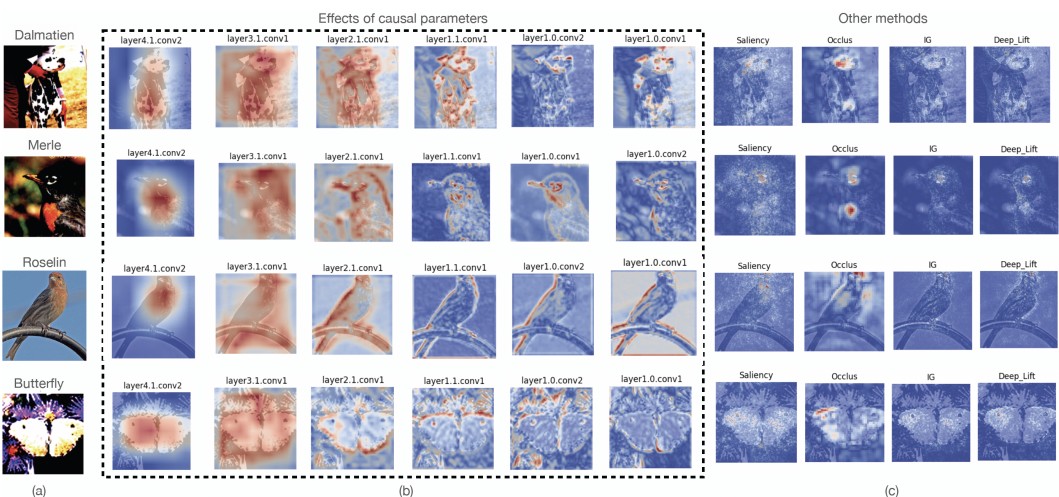

Figure 7: Visualizations and qualitative comparisons. (a) Examples of ImageNet classes. (b) Visual effects (explanations) of causal parameters of some hidden layers of ResNet18. (c) Saliency maps obtained by attribution-based methods.

cision boundary around a local data input with linear classifiers (Ribeiro et al. (2016); Montavon et al. (2017)). Saliency maps (Simonyan et al. (2014); Binder et al. (2016); Selvaraju et al. (2017a); Shrikumar et al. (2017)) are attribution-based methods that describe the relevance of local feature input (e.g., image pixels) on model's output. The explanations of input-output relation for a post-hoc model are generated by one of two different techniques. A backproagation, which rely on class gradients (Selvaraju et al. (2017a); Sundararajan et al. (2017)), and perturbations technique, which use random counterfactual instances (Burns et al. (2020)) or partial substitution of local features (Zeiler & Fergus (2014)) to capture the relevance of such variations on prediction. Although, these methods are significant contributions in DNN interpretability, the explanations can be summerized by finding a correlation between input features or concepts and model output. We believe that robust explanations for complex DNN models require the language of causal reasoning and cause-effect relations.

**Causality for Black Box Explanations.** Structural causal models (SCMs) (Pearl (2009); Peters et al. (2017)) have been recently proposed for extracting local explanations from post-hoc models. These methods focus on local input-output relations by extracting causal explanations from interventions applied on feature space of input data (Chattopadhyay et al. (2019); Schwab & Karlen), or

on the latent space of generative models (Harradon et al. (2018); Goyal et al. (2019); O'Shaughnessy et al. (2020)). Our work is different as we focus on the model itself as a mechanism to capture global and local explanations from the pieces of knowledge encoded in its parameter space during the phase of training.A prior work (Narendra et al. (2018)) has used features importance as a way to extract most and least important filters in a CNN. They used linear regression to transform the vector-valued features to real-valued ones on which interventions where applied to estimate the effect of variance of filters response. Compared to them, our method is fundamentally different. It is not limited to CNN and does not rely on features transformations to extract causal filters as this is highly sensitive to noise and confounding errors. We focus on interventions on the parameter space, and provide a fundamental framework on searching global and local mechanisms to explain the behaviour of post-hoc DNN models.

## 5 CONCLUSION AND DISCUSSIONS

Causality has become an important player in the deep learning field and more particularly, in aspect related to explainability, robustness fairness and bias (Kusner et al. (2017); Kilbertus et al. (2017); Zhang et al. (2021)). In this work, we focused on DNN explanations. We introduced a novel method to capture explanations for post-hoc DNN models which quantify cause-effect relations between model prediction and its parameters. We started our method by constructing the causal diagram that generally describes a post-hoc model. The proposed causal graph is generic as it can be used to capture the effects of interventions on features space, or on parameters space. We then proposed a new algorithm (interventional black box explanations) to capture the effect of parameters on output both locally and globally. Our method is model-agnostic because it does not require specific architecture. We demonstrated that it works on simple architectures, such as LeNet, and complex one including skip connections between structured layers (blocks). Although, in this work we focused on classification architectures, our method can be easily generalized to other architectures like object detection and NLP models. This will be one of the next steps of our future work.

Recently, work on DNN explanations considered logic forms of explanations (Mu & Andreas (2020)). Logic explanations search relations between concept (semantic) neurons in the DNN to generate natural language expressions for model interpretability. These methods limit their explanations to the last hidden layers of the network because of their expensive computational cost. Other limitations that might affect the robustness of explanations in these methods is the presence of some low quality neurons in the network. Our framework can be used as a basic block for these methods. We can use our approach to first capture the causal filters and the corresponding responses (features), then generate logic explanations that expand to all hidden layers in the network.

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

# A APPENDIX

## A.1 DETAILS FOR PROPOSITION 1

To proof Proposition 1, we show in Fig. 8 a toy example of an MLP network with a 1 hidden layer containing 2 units with variables $u_1$ and $u_2$, one output (a simple linear function) $y$ and one scalar input $x$. We start by the direct effect $(w_1^1, w_2^1) \rightarrow y$. Since we have only one class, then $K = 1$. We consider that $do(w_1^1)$ has effect on $y$ and changed the prediction. Then, a causal path exists between $u_1$ and $y$. Since the intervention values $\alpha = 0$, this leads to

$$y = u_1 \times w_1^1 + u_2 \times w_2^1, \qquad \tilde{y} = u_2 \times w_2^1 \quad (w_1^1 = 0) \tag{3}$$

We also have

$$y = x \times w_1^1 \times w_0^1 + x \times w_2^1 \times w_0^2, \qquad \tilde{y} = x \times w_2^1 \times w_0^2 \tag{4}$$

From equation (4), and the fact that we know $w_1^1$ has a significant effect $e$, we then got

$$e = y - \tilde{y} = x \times w_1^1 \times w_0^1 \tag{5}$$

The important information flows from input $x$ to $y$ through the mediator $u_1$, and $w_0^1$ is the selected intervention variable for the effects of the parameters $l_0$ on $y$. It is easy to see this on an input of two variables $(x_1, x_2)$. In this case the parameters $W_0^1 = (w_0^{11}, w_0^{21})$ are the selected interventions variables.

## A.2 DETAILS FOR PROPOSITION 2

In causal graphs (DAG) (Ay & Polani (2008)), the measure of information flow is defined by the Kullback–Leibler divergence which is related to mutual information, and used to quantify the causal effects between disjoint subsets of nodes. Our case is similar as we search to compare the prediction probability of the post-hoc DNN $P(\boldsymbol{y}|\boldsymbol{X})$ with the probability resulted form interventions on $W$, $P(\boldsymbol{y}|do(W), \boldsymbol{X})$. The output is obtained by feedforward the effects of intervention variables to the output. To capture the amount of information related to the action $do(_i^K)$ in the hidden layer $l_i$, we used Kullback–Leibler divergence which provides the result in (2).

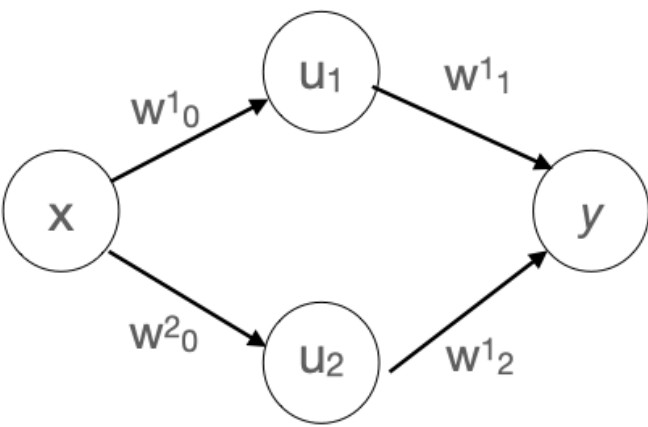

Figure 8: A toy example. MLP with one hidden layer, one input and a linear output

## B APPENDIX

Our method applied to ResNet18 trained on MNIST dataset. The model achieved almost perfect performance on all digits of the MNIST test set. The average performance was $99\%$, so it was rare to find negative samples on which the model wrongly predicted the class. We found only one negative sample corresponding to digit 8, and no negative samples for digit 3. Fig. 9 illustrates explanations computed from two selected convolution layers. We show in a human interpretable way what are the causes for a wrong prediction and what treatment can fix it.

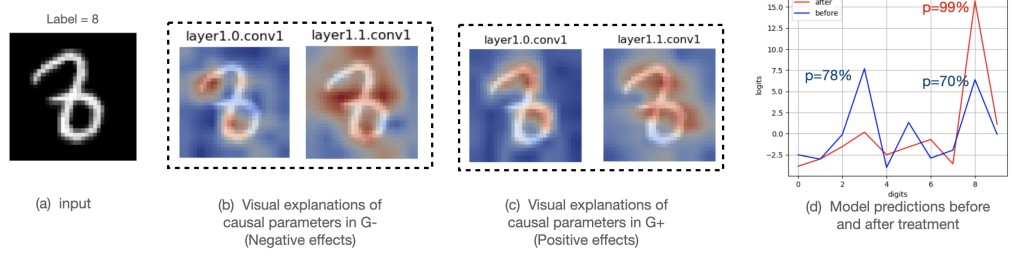

Figure 9: Using causal explanations to correct wrong predictions of ResNet18 for MNIST exmaple.

## C APPENDIX

More visualization details from all hidden convolution layers of ResNet18 on ImageNet classes mentioned in Section 3.3.

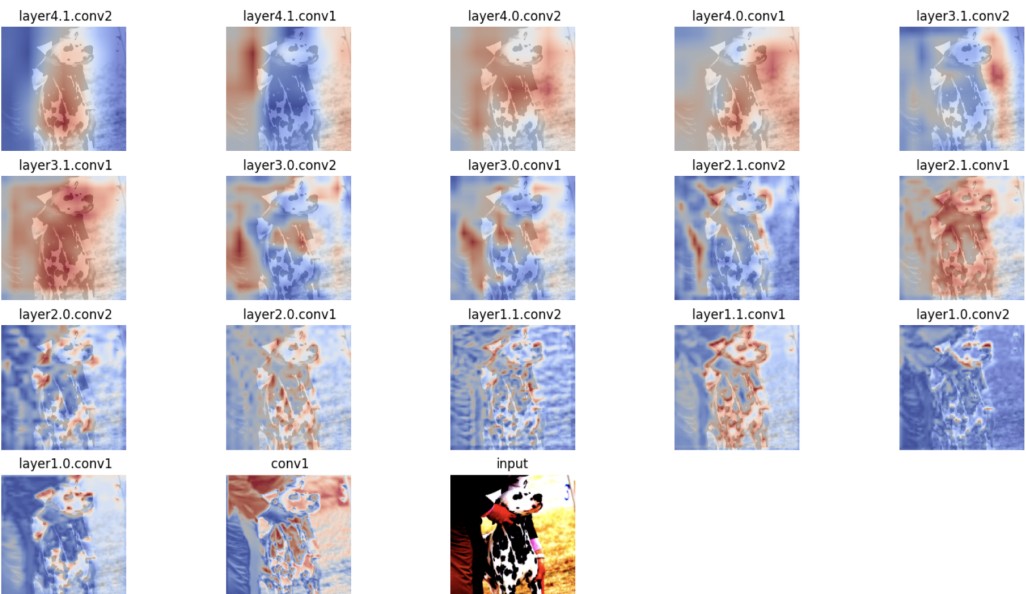

Figure 10: Visualization of causal effects in all convolution layers for the class object Dalmatien.

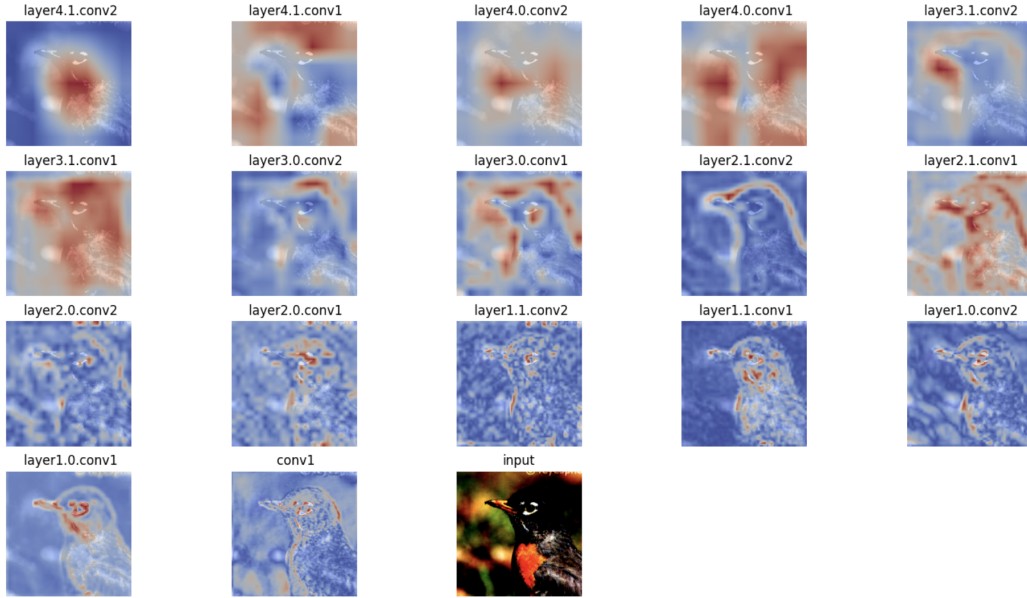

Figure 11: Visualization of causal effects in all convolution layers for the class object Merle.

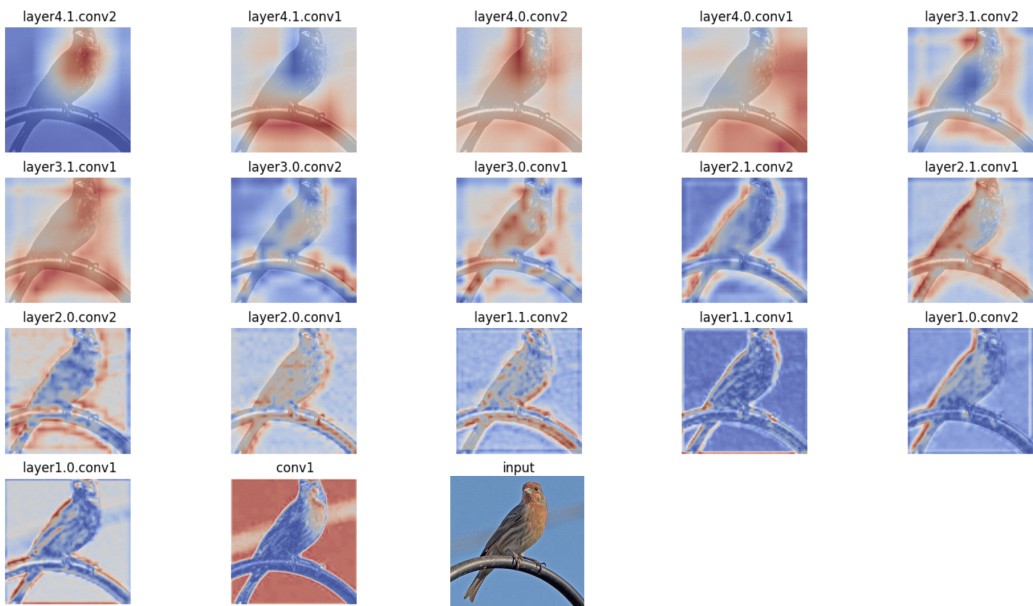

Figure 12: Visualization of causal effects in all convolution layers for the class object Roslein.

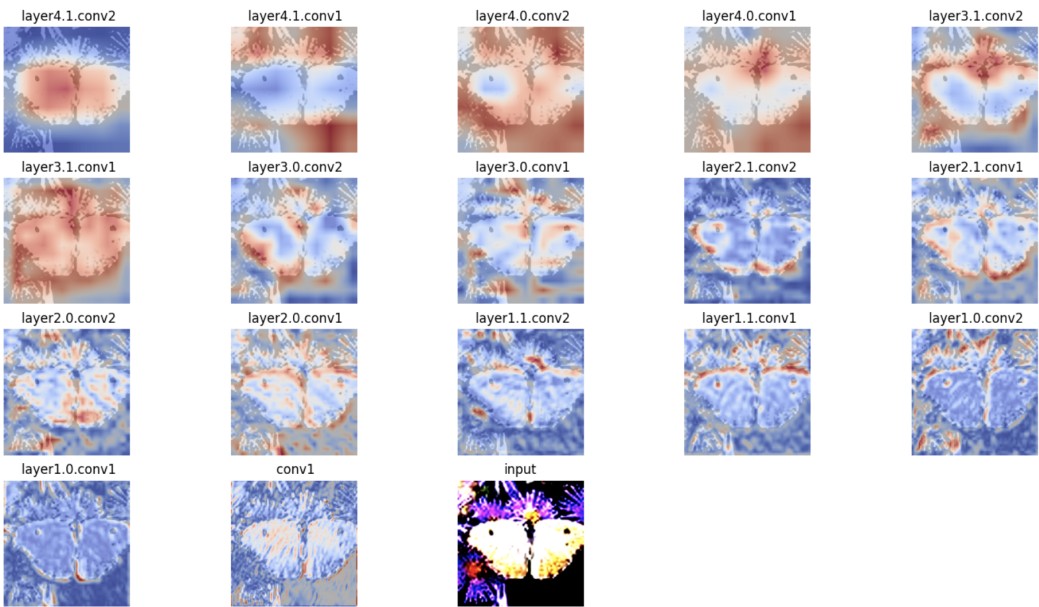

Figure 13: Visualization of causal effects in all convolution layers for the class object Butterfly.

