# OpenReview forum: "Interventional Black-Box Explanations"
_ICLR.cc/2022/Conference — ICLR 2022 Submitted_

### Official Review · Reviewer_TDuY · 2021-10-31

**Correctness:** 2
**Technical Novelty And Significance:** 2
**Empirical Novelty And Significance:** 2
**Recommendation:** 3
**Confidence:** 4

**Main Review:**

Strength:
1. The problem raised by the paper is well-motivated.
2. The visualization of the layer-wise weights is quite intuitive, helps readers to better understand the behaviors of each layer.

Weakness:
Overall, the paper lacks technical rigorous and verbal accuracy which makes the contribution delivered by the authors less convincing.
1. It appears to me that the paper used intervention and do-calculus in an interchangeable manner without clearly defining what 'intervention' means. Under such circumstances, I can only conclude that such language utility is not precise, please check for instance (Peters et al. 2017) for a more precise defintion.

2. In  Assumption 1, the authors claim '... by the measure of changes ... ', what does 'measure of changes' mean? This definition sounds confusing to me.

3. In Proposition 2, the authors name CS_i as the causal effect.  Be aware that there has been a well-established terminology so-called average causal effect used in the causal inference community (See Hernán MA, Robins JM (2020)). The authors should make a clear discussion about how the causal effect defined in the paper related to one that is well-known to the community.

4.  The paper states 'Structural causal models (SCMs) (Pearl (2009); Peters et al. (2017)) have been recently proposed for extracting local explanations from post-hoc models.' Could you be more specific? Correct me if I am wrong, as far as I know, the SCMs proposed in the two books were not meant for trained DNN models. They might be very good candidates for the author's tasks, but that is unclear based on the contents of the two books and remains to be investigated.

5. LeNet and ResNet 18 are less representative compared to the rapid development of the DNN model architectures.


**Summary Of The Paper:**

This paper proposed a causal-driven method aiming to resolve the black-box issue of DNNs. The proposed interventional black-box explanations method tends to be model-agnostic and can apply a variety of DNN models. In the experiments, the authors examined the proposed method on two well-known DNN architectures LeNet and ResNet18.

**Summary Of The Review:**

In weighing the pros and cons of the paper, I believe that the current version is not ready for formal publication.
As well-motivated as the paper can be, a significant revision needs to be done before the paper can bring useful insights to the community.

---

### Official Review · Reviewer_GBET · 2021-11-01

**Correctness:** 3
**Technical Novelty And Significance:** 2
**Empirical Novelty And Significance:** 2
**Recommendation:** 3
**Confidence:** 3

**Main Review:**

I find the overall idea of the paper, i.e. connecting explanations with causal diagrams, quite interesting and promising. This is because it opens an approach to study the underlying mechanisms of the decision making process of the neural network in a principled manner.
I also find the application to use these causal explanations to correct predictions quite intriguing (although I do not find the technical design of the experiments convincing - as discussed below).

Unfortunately, my overall impression is that the manuscript would still require a significant amount of work in order to reach a state in which it could be considered for acceptance. I will try to summarize this in detail below, but before doing so I do want to encourage the authors to invest more time into this work as it is underlying idea is quite interesting.

- I found Section 2.1 a bit chaotic. For example, it contains the sentence "The subscribe number indicates that this is the input layer l_0". It is unclear what this sentence refers to. Another example is that c_i is being defined as the convolution units. This is not standard terminology and one has to guess from the context what is meant. E_{i,i+1} is defined as "the edge vector connecting the nodes in layer l_i to l_{i+1}" but it is again not clearly defined/stated what is meant by this.

- All Figures in the manuscript are very hard to read as they use extremely small font sizes.

- Section 2.2. states that parameters and features form a "collider" but never defines this term. If this is standard terminology in the causal diagram literature then one should add a background section on causal diagrams which contains all relevant definitions. This is particularly important as the main contribution of this work is to introduce these novel techniques to the field of explainability.

- Proposition 1 is only proven for a simple example as far as I can see? This is obviously not enough. If the stated existence of causal paths is assumed axiomatically, this should be made clear. Similarly, Proposition 2 is a definition not a proposition as far as I can tell.

- As intervention, only \alpha_n=0 is used. One may expect that this choice is of the utmost importance for the results. It should thus be motivated and/or compared with other choices.

- Algorithm 1 should clearly state what the input symbols mean, i.e. X input data, y predictions, and so on.

- Per-sample experiments: as far as I understand these experiments were only performed for two selected data samples. If so, it needs to be discussed if they demonstrate generic behavior. It would be preferable to design experiments that are not sensitive to the choice of data sample, for example by reporting averaged values over many data samples.

- Figure 3 does not have a label for the x-axis. Is this the index number? Both Figure 2 and 3 have legends and labels which are significantly too small. It should also be clearly explained what is meant by "conv1 layer". Is this the first convolutional layer (if counted from the input layer upwards)?

- "use explanations for correcting predictions". As far as I understand, the 100% certainty of the model is measured with respect to the samples from which the type of correction is inferred? What happens with the other samples if the same parameters are set to zero?

- Section 3.3 contains broken citation references for DeepLift and Assumption 1.

- Section 3.3 incorrectly states that LRP does not work on ResNet, see e.g. 2003.07631. There are several open source implementations for LRP on various architectures including ResNets, see e.g. https://github.com/chr5tphr/zennit.





**Summary Of The Paper:**

The manuscript proposes an explanation method for feedforward neural networks which is based on causal diagrams.

It evaluates the proposed method using two architectures (LeNet and Resnet18) and two datasets (MNIST and a subset of ImageNet) and performs a comparison to existing explanation methods.

**Summary Of The Review:**

Overall, I think that the underlying idea of the manuscript is promising but that the draft is immature and requires a significant rewrite to more clearly explain the underlying notation, background, and theoretical assumptions. Furthermore, the experiments need to be extended and more clearly explained. Please refer to the list of feedback above for details.

---

### Official Review · Reviewer_Acmr · 2021-11-02

**Correctness:** 3
**Technical Novelty And Significance:** 3
**Empirical Novelty And Significance:** 2
**Recommendation:** 3
**Confidence:** 3

**Main Review:**

The motivation for the paper is clear and well written and the idea is good. There is also relatively little work on explanations from model manipulation, so the direction of the paper is promising. The paper is well structured and the setup is described clearly.

**I do have a few concrete questions and remarks regarding Clarity:**

**In algorithm 1: **
The index of weight $w_l^k$ is added to $I_+$ if $w_l^k$ has a positive effect on the correct class, that is when when $\tilde{y}>y$, then if $\delta_l$ is positive shouldn't it say "Add the index of $w$ to $I_+$ if $\tilde{y} − y > \delta_l$"?

**Figure 2:**
The text in the legend and on the axis is way to small!
Are these averages over several samples of 3 and 8 or just one?

Here it would be interesting to
1. get quantitative graphs
2. compare digits that are similar (3 and 8) AND very different (1 and 5)

The authors state "Fig. 2 demonstrates an evidence on the presence of causal mechanisms that have either positive or negative effects on model prediction."
How does the figure show both negative and positive effects?

**Figure 3:**
The text in the legend and on the axis is way to small!
there is no x-label.
I haven't completely understood what Figure 3 shows and how it relates to figure 1, it would be good to explain this a bit more.

**Figure 4:**
The text in the legend and on the axis is way to small!
Here a scatter plot with 8 on one and 3 on the other axis makes more sense as the colour and shape of the circles already represent frequency.

**Cross-samples causal mechanisms:**
Are the 100 positive samples taken from the trainings set or the test set?

**3.2 use explanations to correct predictions:**
So here the model was pruned and then tested again?
Here it is very important what data was used for pruning. The authors need to explain the process more rigorously.
Also, what does it mean to say "the model is 100% certain of its prediction"? What is the test accuracy of the pruned model?

**3.3 Comparison to explanation methods**
The authors state "We obtain the attribution map, by computing the average at each pixel." It would be advantageous to explain in a bit more detail how they get from a collection of indices of relevant weights to an attribution map.
I think, LRP does exist for ResNet, see "INNvestigate Neural Networks" (Alber et al.).
It is not obvious to me how the authors conclude "Our method shows the effects of the causal filters which explain the difference between digits through their shapes."

**A.1**
The proposed MLP seems to have no activation functions and is therefor just a one layer linear mapping which can be reduced to y=x*w. It is unclear what the authors intend to show with this.
The notation is not clear. What are $w_0^{11}$ and $w_0^{21}$?

**B and C**
How can we interpret the attribution maps?


**I also have a few concerns regarding quality of the paper:**

- The attribution maps shown are not very convincing and it's unclear in which ways they provide better insights than existing explanation methods.

- There is no quantitative evaluation of explanations for example with ROAR (2019, Hooker et al.).

- Figure legends and axis labels are too tiny to be read.

- The paper "Interpreting CNN Knowledge Via An Explanatory Graph" by Zhang et al. seems very closely related but is not discussed in the current paper.

- The authors do hint on time constraints when applying the do- operation to all parameters of a model but do not give further insight into how many parameters need to be tested on average and how long this taked for bigger models such as ResNet

**Summary Of The Paper:**

The authors aim to give post hoc explanations of Neural Network classifier decisions. They do so by finding causal relationships between model parameters and classifier outputs. To this end model parameters are set to zero and the change in the models prediction is calculated. If the change in prediction exceeds a threshold the parameter is deemed relevant and used for construction of an attribution map.

**Summary Of The Review:**

There is relatively little research on Explanation by manipulating weights. So the direction of the paper is interesting and relevant.
Unfortunately the paper is lacking clarity in many aspects, for example how attribution maps are extracted or the description of experiments, especially the "model treatment".
It is unclear how the attribution maps give better insight in model behaviour than other explanation methods.
There is no quantitative evaluation of results.

---

### Official Review · Reviewer_NpLg · 2021-11-03

**Correctness:** 3
**Technical Novelty And Significance:** 2
**Empirical Novelty And Significance:** 2
**Recommendation:** 3
**Confidence:** 3

**Main Review:**

In general, I found the paper hard to read, with notation and terminology rather confusing. Below are a few examples:
- The definition of w_i^k could be a lot clearer, for example, explaining that the intervention involves setting multiple *edges* that lead to a neuron to zero, and perhaps drawing a connection to dropout where neuron weights are zeroed out.
- I think the terminology “A de-confounded (robust) explanation of a black-box”. As far as I can tell, this is the only use of the term de-confounded in the paper; it seems unnecessary. What is being proposed here also seems less of an assumption and more like something you are defining in this paper.
- The \tilde{y} notation is a little confusing. On page 4 it is mentioned that mu_{\tilde{y}} is the average value of the changed prediction, and \tilde{y} next appears in Algorithm 1 where it is used in a difference term: \tilde{y} - y.
- The terminology “post-hoc DNN”” is unusual. For example, “We are interested in explaining post-hoc models, i.e., models after training”, “We build a causal graph of a post-hoc DNN”, “In post-hoc DNNs, each path communicates the information”. IMO, explanations can be called post-hoc explanations but the DNN model should not be called post-hoc.

I think it should be clarified that the interventions are not independent (i.e. different interventions w_i^k could be operating on some common set of edges), and that the earlier layers are ancestors of the later layers. I am not sure how to reconcile this with Algorithm 1 running from the penultimate layer to the initial layers.

I have a few questions about the experimental results:
- In Figure 7, with the exception of Occlus, Saliency, IG, Deep_Lift all look very blue. Could this be due to a scaling issue? Otherwise, these results suggest that IG and DeepLift methods found nothing at all important in these images, which is seems unusual.
- Figures 2-5 are hard to read — x and y axes labels could be larger — and I found it hard to interpret Figure 2 and 3.

Citations:
- (Page 7) Citation for DeepLift is missing
- (Page 7) Letham 2015 is cited in the context of global approximations approximating DNNs but this paper proposes a ground-up interpretable model, bayesian rule lists, and doesn’t have DNNs

Typos:
- (Page 3) “The graph allows us isolating…” —> The graph allows us to isolate
- (Page 6) “Based on Assumption ??”
- (Page 8) “A backproagation”
- (Page 9) “and more particularly, in aspect related to” —> in aspects related to

**Summary Of The Paper:**

The paper observes that many DNNs are DAGs and proposes to run interventions on parts of the network then compute the effect of the interventions, which the paper calls causal explanations of the DNNs.


**Summary Of The Review:**

The idea is interesting but the terminology and experiment results are confusing.

---

### Author Response · Authors · 2021-11-09
**General response to all reviewers**

We thank the reviewers for their time and comments on the paper. All the reviewers appreciated the idea of our paper and agreed on the lack of rigorous details and sufficient evaluations. With respect to the reviewers evaluations, we decided to withdraw our paper and invest more time on improving our work taking into account reviewers comments and questions.

---

### Decision · Program_Chairs · 2022-01-20

**Decision:**

Reject

**Comment:**

The reviewers are in consensus. I recommend that the authors take their recommendations into consideration in revising their manuscript.